# Optical-helicity-driven magnetization dynamics in metallic ferromagnets

Gyung-Min Choi[1,2], André Schleife[2] & David G. Cahill[2]

Recent observations of switching of magnetic domains in ferromagnetic metals by circularly polarized light, so-called all-optical helicity dependent switching, has renewed interest in the physics that governs the interactions between the angular momentum of photons and the magnetic order parameter of materials. Here we use time-resolved-vectorial measurements of magnetization dynamics of thin layers of Fe, Ni and Co driven by picosecond duration pulses of circularly polarized light. We decompose the torques that drive the magnetization into field-like and spin-transfer components that we attribute to the inverse Faraday effect and optical spin-transfer torque, respectively. The inverse Faraday effect is approximately the same in Fe, Ni and Co, but the optical spin-transfer torque is strongly enhanced by adding a Pt capping layer. Our work provides quantitative data for testing theories of light–material interactions in metallic ferromagnets and multilayers.

[1] Center for Spintronics, Korea Institute of Science and Technology, Seoul 02792, Korea. [2] Department of Materials Science and Engineering and Materials Research Laboratory, University of Illinois, Urbana, Illinois 61801, USA. Correspondence and requests for materials should be addressed to G.-M.C. (email: gm_choi@kist.re.kr) or to D.G.C. (email: d-cahill@illinois.edu).

Manipulation of magnetization via light is a key aspect of ultrafast spintronics. Beaurepaire et al.[1] demonstrated that photon energy can be transferred to magnetization on a femtosecond time scale in a metallic ferromagnet. Later, Stanciu et al. showed that circular polarization of light can switch magnetization of a metallic ferrimagnet without the use of a magnetic field[2]. These results have led to the emerging field of all-optical helicity-dependent switching (AO-HDS)[2–7]. Until recently, AO-HDS was confined to ferrimagnetic systems, in which two sublattices are antiferromagnetically coupled; the mechanism of switching is connected to the compensation temperature, where the magnetizations of sublattices sum to zero[8–10]. Based on this understanding, however, the recent observation of AO-HDS in metallic ferromagnets was unexpected[11]. Helicity-dependent switching require a mechanism for angular momentum transfer from light to magnetization, but the mechanism for metallic ferromagnets is unknown.

The direct excitation of spin populations using light has been investigated primarily in semiconductors[12–17]. Circularly polarized light can generate spin-polarized electrons in the conduction band due to the optical selection rules for dipole transitions[12–17]; this mechanism, often referred to by the term 'optical orientation', has practical importance for the generation of spin-polarized electron beams by photocathodes. In a semiconductor that has a net magnetic moment, spin-polarized electrons can interact with the magnetization of the semiconductor via optical-spin-transfer-torque (OSTT)[18,19]. OSTT in a semiconductor is the combination of two well-known mechanisms: optical orientation[12–17] and spin-transfer torque between spin-polarized electrons and local magnetization[20–22].

An additional mechanism for optical helicity-driven magnetization dynamics is the inverse Faraday effect (IFE). IFE was first discovered in insulating paramagnets[23–25]; IFE-driven magnetization dynamics has been reported in experiments on both insulating[26,27] and metallic ferrimagnets[28]; recently, IFE was invoked to explain terahertz-frequency emission by metallic ferromagnets[29]. Although IFE has been proposed as the mechanism for AO-HDS for ferromagnetic metals[11], a rigorous theory for IFE in metals is still under development[30–36]. The original theory of refs 24,25 developed for insulators explains IFE in terms of an interaction Hamiltonian that couples angular momentum of light and material; this interaction Hamiltonian induces an optomagnetic field which is proportional to $|E|^2$, where $E$ is the electric field of light inside the material. This optomagnetic field produces a field-like torque that drives magnetization dynamics.

Theories of IFE are sometimes based on induced magnetization rather than an optomagnetic field[32,34]. To clarify the discussion, we note key differences between the IFE-driven and OSTT-driven magnetization. First, the IFE-induced magnetization is derived from a second-order perturbation with respect to $E$-field of light[32,34], while the OSTT-driven magnetization is derived from a first-order perturbation[14,17]. Second, the relationship between magnetization, $m$, and light intensity, $I$, is different: $I \propto m$ for IFE; $I \propto (dm/dt)$ for OSTT. In other words, the IFE-induced magnetization is an equilibrium quantity calculated from the second-order density matrix response[32,34], while the OSTT-induced magnetization is derived from the rate of spin generation, calculated from probability of interband transitions[14,17]. The IFE-induced $B$-field and magnetization can be related by $m = \chi_m B$, where $\chi_m$ is the static magnetic susceptibility[32]. However, for the IFE-driven magnetization by short optical pulse, we must consider dynamic behaviour. For example, the alignment of magnetization along the $B$-field will take a few nanoseconds for Co as magnetization undergoes a damped precessional motion to approach the equilibrium position. However, if the alignment of magnetization occurs during the pulse duration, we can treat IFE as a magnetization rather than a $B$-field[32,34]. In our analysis, we assume that the timescale for IFE to induce magnetization is much longer than the pulse duration, and treat IFE as a transient $B$-field created by the optical pulse and solve the torque equation.

Our experiments are designed to understand the magnitude and mechanisms of the optical-helicity-driven magnetization dynamics in metallic ferromagnets (FM): Co, Fe and Ni. We observe that circular polarized light produces magnetization dynamics that is explained by a combination of a spin-transfer torque that we attribute to spin polarization generated by OSTT and a field-like torque that we attribute to optomagnetic field generated by IFE. With a thin Au or MgO capping layer on top of the FM layer, the field-like torque dominates the magnetization dynamics. Replacing the Au or MgO capping layer with a Pt capping layer results in a significant enhancement of the spin-transfer torque, while the field-like torque remains approximately constant.

## Results

**Sample preparation and optical measurement.** The film structure that we study is sapphire substrate/FM(10)/capping(x), where FM is Co, Fe or Ni with thickness of 10 nm, and capping(x) is Au, MgO or Pt with thickness of $x$ nm. All layers are deposited by magnetron sputtering with base pressure of $< 5 \times 10^{-8}$ Torr. For optical measurements, we use a time-resolved pump-probe technique (see Methods). The circularly polarized pump light is incident on the substrate side of the samples. A photon with left circular polarization (LCP) and right circular polarization (RCP) carries a spin angular momentum of $+\hbar$ and $-\hbar$, respectively. (For light helicity, we adapt the view point from the receiver, which is the most common convention in optics[37].) The linearly polarized probe light is incident on the surface side of the samples and detects the magnetization dynamics by magneto-optical Kerr effect (MOKE). We use high frequency modulation and balanced detection to minimize noise level (see Methods). All experiments are performed at room temperature.

**Two orthogonal torques by optical pulse.** Figure 1 illustrates our assignments of two orthogonal torques to helicity-driven magnetization dynamics: the spin-transfer torque by OSTT and the field-like torque by IFE. Circularly-polarized pump light is incident on a FM thin film in the $z$-direction; the magnetization of FM lies in the $x$-direction. The magnetization dynamics driven by OSTT and IFE during the pump pulse can be expressed as[18,20],

$$\frac{d\widehat{\mathbf{M}}}{dt} = -\frac{1}{M}\widehat{\mathbf{M}} \times \left(\widehat{\mathbf{M}} \times \frac{d\mathbf{m}_{sp}}{dt}\right) - \gamma\widehat{\mathbf{M}} \times \mathbf{B}_{opt}, \qquad (1)$$

where $\widehat{\mathbf{M}}$ is the unit vector of magnetization of FM, $M$ is the magnitude of magnetization of FM, $\mathbf{m}_{sp}$ is the OSTT-driven spin polarization, $\gamma$ is the gyromagnetic ratio and $\mathbf{B}_{opt}$ is the IFE-driven optomagnetic field. The $\mathbf{m}_{sp}$ applies torque to the $z$-direction, while the $\mathbf{B}_{opt}$ applies torque to the $y$-direction. When magnetization is tilted from the equilibrium position by transient torque during the optical pulse, its dynamics after the optical pulse is governed by the effective $B$-field, determined by shape anisotropy, crystalline anisotropy and external magnetic field. Since the effective $B$-field is along the $x$-direction, the magnetization dynamics is a damped precessional motion in the $y$–$z$ plane with the centre axis in the $x$-direction.

**Time-resolved MOKE measurement.** We measure the $z$-component of magnetization ($M_z$) dynamics of the Co(10)/Au(2) sample using polar MOKE. Pump light triggers

precession of magnetization, and this magnetization dynamics changes sign with pump helicity (Fig. 2a). A small helicity-independent (asymmetric) component of the magnetization dynamics is due to a small misalignment between the orientations of the crystalline anisotropy and the applied magnetic field (Supplementary Note 1). We extract the helicity-dependent (symmetric) component by taking the difference $(\Delta\theta_{L-R})$ between LCP and RCP (Fig. 2b). We fit the data for $\Delta\theta_{L-R}$ with a damped cosine function of the form $\cos(2\pi ft - \phi)\exp(-t/\tau)$ where $f$ is the precession frequency, $f = 8.5$ GHz, $t$ is the time delay between pump and probe, $\phi = 65°$ is the phase delay and $\tau = 600$ ps is the time constant for exponential decay. Later, we convert $\Delta\theta_{L-R}$ to the relative change of magnetization $(\Delta M/M)$ using $(\Delta M/M) = (\Delta\theta_{LR}/2\theta_K)$, where $\theta_K$ is the Kerr rotation angle corresponding to saturation magnetization (see Table 1). The parameters $f$ and $\tau$ are determined by saturation magnetization and damping constant of FM, respectively (Supplementary Note 2).

The phase delay, $\phi$, of $M_z$ dynamics is determined by the direction of the initial tilting of magnetization. When the initial tilting of magnetization is along the $z$-direction, $\phi$ should be 0° (Fig. 2 of ref. 18). When the initial tilting of magnetization is

along the $y$-direction, $\phi$ should be 90° (Fig. 2 of ref. 27). The large $\phi$ of 65° suggests that the initial tilting of magnetization is closer to the $y$-direction than $z$-direction. We also observe a large phase delay in the $M_z$ dynamics in Fe and Ni, (Fig. 3), and when a different capping layer, MgO, is used (Supplementary Note 3).

To obtain a more complete picture of the magnetization dynamics, we measure the $y$-component of magnetization $(M_y)$ dynamics using longitudinal MOKE in the $y$–$z$ plane (Fig. 3). $M_y$ dynamics is 90° out of phase with $M_z$ dynamics and has $4\sim6$ times larger amplitude than $M_z$ dynamics due to the shape anisotropy of FM. We also measure the $x$-component of magnetization dynamics using longitudinal MOKE in the $x$–$z$ plane, but found no helicity dependence. Although energy of light induces ultrafast demagnetization in the $x$-direction, there is no optical-helicity dependence on $M_x$ dynamics (Supplementary Note 4).

**Effect of a Pt capping layer**. Replacing the Au capping layer with a Pt capping layer decreases phase delay and increases the amplitude of $M_z$ dynamics (Fig. 3). Especially with Co(10)/Pt(4) samples, the amplitude of precession is about four times larger than that of Co(10)/Au(2) sample, and the phase delay decreases to $\sim10°$. From optical calculation, the Pt layer absorbs 18% and 30% of light energy absorbed in the Co(10)/Pt(2) and Co(10)/Pt(4) structure, respectively; the Au layer absorbs 1% of light energy absorbed in the Co(10)/Au(2) structure (Supplementary Note 5).

We determine the initial $M_z$ and $M_y$ tilting immediately after the pump pulse ($t = 1$ ps) by analysing the amplitude and phase of $M_z$ and $M_y$ dynamics shown in Fig. 3. The initial $M_y/M$ tilting is insensitive to the capping layers: $-2\times10^{-4}$ for Co and Fe and $-3\times10^{-4}$ for Ni samples with LCP pump (the sign changes with RCP). However, the initial $M_z/M$ tilting is highly sensitive to the capping layers: the initial $M_z/M$ tilting increases from $-0.2\times10^{-4}$ for Co(10)/Au(2) sample to $-1.7\times10^{-4}$ for Co(10)/Pt(4) sample with LCP pump (the sign changes with RCP). The similar enhancement of the $M_z/M$ tilting is also observed in Fe and Ni samples (Fig. 4).

**Quantification of IFE**. From the initial $M_y$, we evaluate the optomagnetic field needed to generate the field-like torque. The magnetization tilting along the $y$-direction generated by the pump pulse is related to the optomagnetic field by

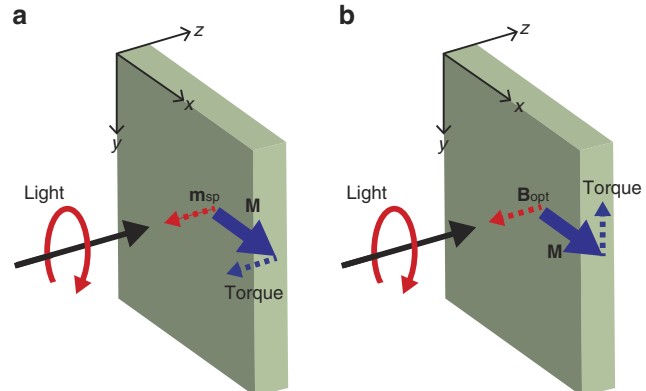

**Figure 1 | Schematic representation of two orthogonal torques.** Circularly polarized light, incident on magnetization of ferromagnet (**M**), generates either spin polarization (**m**$_{sp}$) via optical orientation or optomagnetic field (**B**$_{opt}$) via inverse Faraday effect. (Black arrow indicates a wavevector of light along the $z$-direction, and red circular arrow indicates left-circular polarization.) (**a**) The **m**$_{sp}$ (red dotted arrow) rotates **M** (blue solid arrow) to the $z$-direction by spin-transfer torque (blue dotted arrow). (**b**) The **B**$_{opt}$ (red dotted arrow) rotates **M** (blue solid arrow) to the $y$-direction by field-like torque (blue dotted arrow).

$$\frac{M_y}{M} = \gamma \int B_{opt}(t)dt = \gamma B_{opt} t_{pulse}, \qquad (2)$$

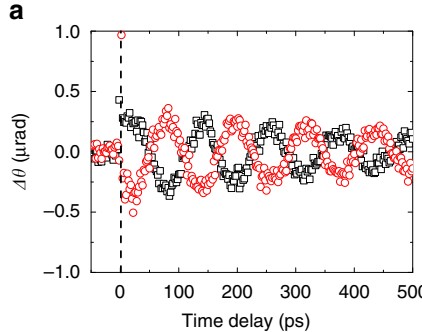

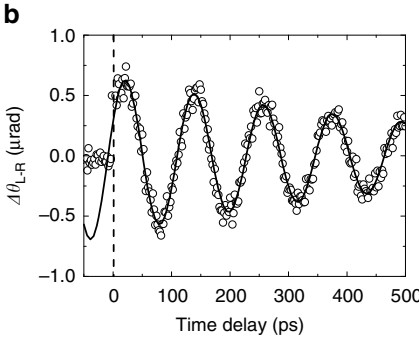

**Figure 2 | Polar MOKE result of the Co(10)/Au(2) sample.** (**a**) Data with left circularly polarized (LCP) (black squares) and right circularly polarized (RCP) (red circles) pump. (**b**) The helicity-dependent magnetization dynamics obtained by subtraction of the data with LCP and RCP. The black line is the damped cosine function of $A\cos(2\pi ft - \phi)\exp(-t/\tau)$, where $A = 0.65$ μrad, $\phi = 65°$ and $\tau = 600$ ps. The dashed vertical line indicates time delay of 1 ps. All measurements are done with incident pump fluence of 10 J m$^{-2}$.

**Table 1 | Amplitude and phase of $M_z$ precession of all samples.**

|  | Co(10)/Au(2) | Co(10)/Pt(2) | Co(10)/Pt(4) | Fe(10)/Au(2) | Fe(10)/Pt(2) | Ni(10)/Au(2) | Ni(10)/Pt(2) |
|---|---|---|---|---|---|---|---|
| $\Delta\theta_{L-R}$ (μrad) | $0.7 \pm 0.2$ | $1.6 \pm 0.3$ | $2.3 \pm 0.5$ | $0.5 \pm 0.1$ | $1.3 \pm 0.3$ | $0.8 \pm 0.2$ | $1.4 \pm 0.3$ |
| $\theta_K$ (mrad) | $-8.0 \pm 0.8$ | $-7.8 \pm 0.8$ | $-6.9 \pm 0.7$ | $-6.5 \pm 0.7$ | $-6.5 \pm 0.7$ | $-3.3 \pm 0.3$ | $-3.2 \pm 0.3$ |
| $\frac{M_z}{M}\big|_{amp}$ $(10^{-4})$ | $-0.4 \pm 0.1$ | $-1.0 \pm 0.3$ | $-1.7 \pm 0.5$ | $-0.4 \pm 0.1$ | $-1.0 \pm 0.3$ | $-1.2 \pm 0.4$ | $-2.2 \pm 0.6$ |
| $\phi$ (°) | $65 \pm 5$ | $22 \pm 5$ | $12 \pm 5$ | $63 \pm 5$ | $19 \pm 5$ | $45 \pm 5$ | $20 \pm 5$ |

The $\Delta\theta_{L-R}$ is the peak Kerr rotation due to the optical-helicity-dependent part of the magnetization dynamics. The $\theta_K$ is the static Kerr rotation due to full magnetization. The $\frac{M_z}{M}\big|_{amp}$ and $\phi$ are amplitude and phase of $M_z$ precession, respectively.

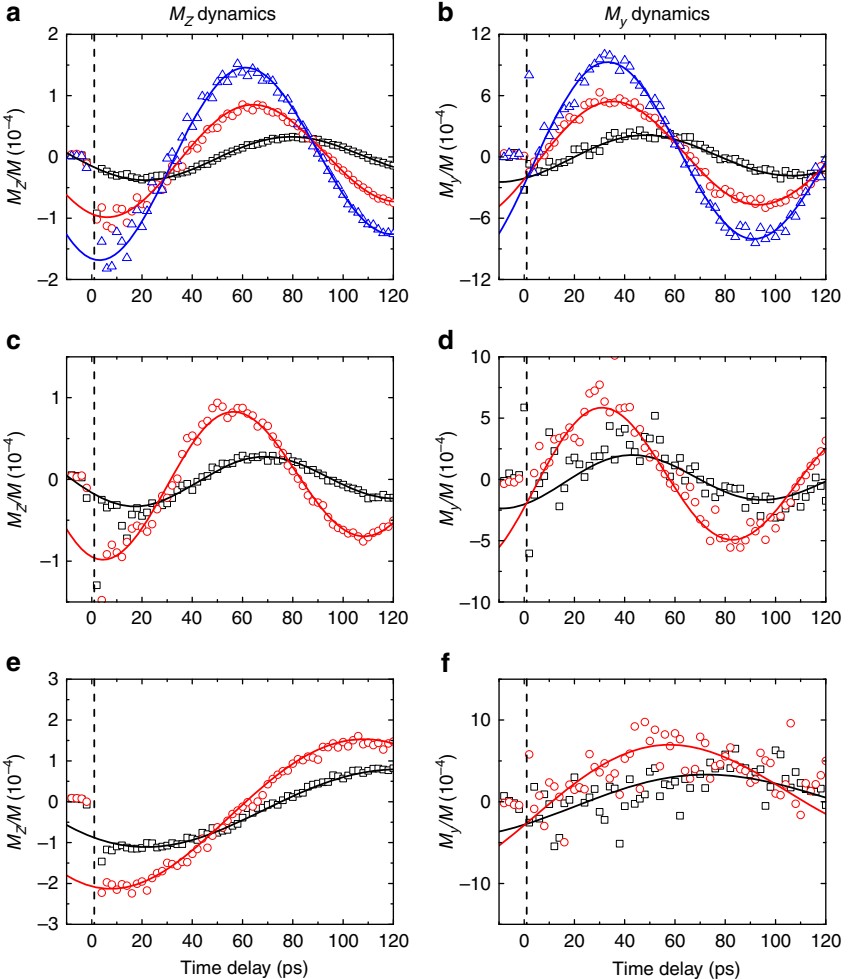

**Figure 3 | Helicity-driven $M_z$ and $M_y$ dynamics of all samples.** The different sample capping layers are Au(2) (black squares), Pt(2) (red circles) and Pt(4) (blue triangles). (**a,b**) Data for Co(10)/capping. (**c,d**) Data for Fe(10)/capping. (**e,f**) Data for Ni(10)/capping.

where $B_{opt}$ is the optomagnetic field averaged over pulse duration, $t_{pulse} = 1.1$ ps is the pulse duration of pump. $M_y/M = -2 \times 10^{-4}$ corresponds to $B_{opt} = 1$ mT along the negative $z$-direction. Considering the $|E|^2 \approx 10^{15}$ V$^2$ m$^{-2}$ inside FM (Supplementary Note 6), where $E$ is the amplitude of electric field of light, $B_{opt}/|E|^2 \approx 10^{-18}$ T m$^2$ V$^{-2}$. This value of the optomagnetic field is close to the value derived from a study of an insulating ferrimagnet[27] (Supplementary Note 7).

According to theory for transparent materials, the optomagnetic field generated by IFE can be related to the Faraday effect[38]. Faraday rotation ($\theta_F$) is due to the difference in the real part of refractive index ($n_{L/R}$) of LCP and RCP and increases linearly with the path length ($l$) of light inside materials as $\theta_F = -(\omega l/2c)(n_L - n_R)$, where $\omega$ is the angular

frequency of light, and $c$ is the speed of light in vacuum. (We adopt the sign convention from ref. 39.) Adopting the view point that helicity-dependent refractive indexes are a result of helicity-dependent resonant frequencies of electron ($\omega_{L/R}$) (ref. 38), the Hamiltonian for Faraday effect can be described as

$$H_F = \frac{\hbar(\omega_L - \omega_R)}{2} = \frac{\hbar(n_L - n_R)}{2}\frac{d\omega}{dn} = -\frac{\hbar c\theta_F}{\omega l}\frac{d\omega}{dn}, \quad (3)$$

where $\theta_F/l$ is $6.3 \times 10^5$, $6.1 \times 10^5$ and $1.7 \times 10^5$ rad m$^{-1}$ for Co, Fe and Ni, respectively at wavelength of 830 nm (ref. 40) (values at wavelength 784 nm have not been reported), $\hbar(d\omega/dn)$ is $-1$ eV for Co and Ni and $-2$ eV for Fe at wavelength of 784 nm (ref. 41). If we assume that $H_F$ is responsible for IFE as well, the IFE-driven $B_{opt}$ can be expressed as $B_{opt} = -(H_F n_{pN}/M)$, where

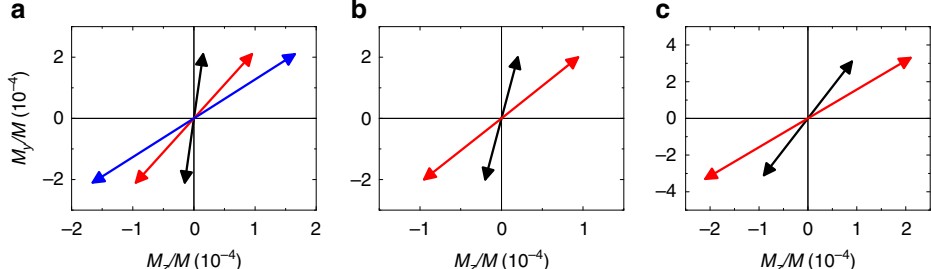

**Figure 4 | Helicity-driven $M_z$ and $M_y$ tilting at $t = 1$ ps.** (**a**) Determination of Co(10)/capping. (**b**) Determination of Fe(10)/capping. (**c**) Determination of Ni(10)/capping. Capping layers are Au(2) (black arrows), Pt(2) (red arrows) and Pt(4) (blue arrows). Arrows with negative/positive $M_z$ and $M_y$ are for left/right circularly polarized pumping.

$n_{pN}$ is the photon density inside the material. The $n_{pN}$ can be obtained from $|E|^2$ by $n_{pN} = (\langle u \rangle / \hbar \omega) = (\varepsilon_0 n^2 |E|^2 / 2\hbar \omega)$, where $\langle u \rangle$ is the time-averaged energy density of light inside FM, and $\varepsilon_0$ is the vacuum permittivity. Then the intensity-normalized $B_{opt}$ is,

$$\frac{B_{opt}}{|E|^2} = \frac{c\theta_F \varepsilon_0 n^2}{2l\omega^2 M} \frac{d\omega}{dn}, \qquad (4)$$

which is calculated to be $1 \times 10^{-18}$ T m$^2$ V$^{-2}$ for Co and Ni and $2 \times 10^{-18}$ T m$^2$ V$^{-2}$ for Fe along the negative $z$-direction for LCP. These estimates are consistent with our experimental results and suggest that the Hamiltonian responsible for Faraday effect is a useful approximation for IFE.

We compare our result to reported IFE theories for metals. Mondal and co-workers estimated $B_{opt}/|E|^2 \approx (1 + \chi_e)10^{-22}$ T m$^2$ V$^{-2}$, where $\chi_e$ is the electrical susceptibility[33]. Using the relation, $\varepsilon_r = 1 + \chi_e$, where $\varepsilon_r$ is the relative permittivity, and $\varepsilon_r$ of $-16.5 + i\,23.3$ for Co at wavelength of 784 nm (ref. 41), and $|E|^2 \approx 10^{15}$ V$^2$ m$^{-2}$ in our case, $B_{opt} \approx 3 \times 10^{-6}$ T. The authors of ref. 33 raised a possibility of much larger $\chi_e$ for Fe relating $\chi_e$ to the anomalous Hall coefficient. Berritta and co-workers calculate the IFE-induced magnetization in metallic ferromagnets and related it to $B_{opt}$ (ref. 34). Converting their calculation with the $I_0 = 10^{14}$ W m$^{-2}$ to our case with $I_0 \approx 10^{13}$ W m$^{-2}$, $B_{opt} \approx 2 \sim 40$ T. Note that there is asymmetry for LCP and RCP because magnetization direction and light propagation direction is the same in their case while in our experiments the magnetization is orthogonal to the direction of light propagation. Qaiumzadeh and co-workers calculate $B_{opt}$ in metallic ferromagnets from the direct optical transition of spin-split sub-bands[35]. Since magnetization and light propagation lie to the same direction, their results also show asymmetry for LCP and RCP. Converting their calculation with the $E_0 = 10^9$ V m$^{-1}$ to our case with $E_0 \approx 10^8$ V m$^{-1}$, $B_{opt} \approx 0.02 \sim 0.2$ T. Freimuth and co-workers consider both IFE and OSTT effects on metallic ferromagnets[36]. They show that both IFE and OSTT depend on quasiparticle broadening ($\Gamma$) and spin–orbit interaction. With a $\Gamma = 25$ meV for room temperature, they predict $B_{opt}$ of 20 and 1.5 mT for Co and Fe, respectively, at $I_0 \approx 10^{13}$ W m$^{-2}$. The result for Fe is close to our experiment but not for Co.

**Quantification of OSTT.** From the initial $M_z$, we evaluate the spin polarization for OSTT. When the light-induced spin polarization is fully absorbed by magnetization of ferromagnet, the $M_z$ tilting of ferromagnet by pump pulse is directly related to the spin polarization by,

$$\frac{M_z}{M} = \frac{1}{M} \int \frac{dm_{sp}}{dt}(t)dt = \frac{m_{sp}}{M}, \qquad (5)$$

where $m_{sp}$ is the spin polarization integrated over the pulse duration. From $M_z/M$, the $m_{sp}$ is 30, 130 and 230 A m$^{-1}$ for Co(10)/Au(2), Co(10)/Pt(2) and Co(10)/Pt(4), respectively. Considering the light absorption in Pt (18% for Co(10)/Pt(2) and 30% for Co(10)/Pt(4)), we conclude that much larger $m_{sp}$ is coming from Pt than from Co.

We use these data to determine the degree of spin polarization (DSP), a parameter often used in discussions of optical orientation in semiconductors. DSP is the ratio between spin polarized electrons ($n_s$) and total electrons ($n_{tot}$) excited by dipole transitions. Since $n_s$ is related to $m_{sp}$, and $n_{tot}$ is related to the amount of light absorption, DSP can be derived from,

$$\mathrm{DSP} = \frac{n_s}{n_{tot}} = \frac{m_{sp}d}{\mu_B} \frac{\hbar \omega}{F_{abs}P} \qquad (6)$$

where $d = 10$ nm is the thickness of the Co layer, $\mu_B$ is the Bohr magneton, $F_{abs} = 6$ J m$^{-2}$ is the total absorbed fluence (determined from $F_{abs} = F_{in}(1 - R - T)$, where $F_{in}$ is incident pump fluence of 10 J m$^{-2}$, $R$ is reflectance and $T$ is transmittance), $P = 0.3$ is the percentage of light absorption of Pt in the Co(10)/Pt(4), and $\hbar \omega = 1.58$ eV is the photon energy. The difference of $m_{sp}$ between Co(10)/Pt(4) and C(10)/Au(2) samples leads to DSP $\approx 0.03$ for Pt. (We estimate DSP of Co, Fe and Ni is at least one order of magnitude smaller than that of Pt.) Note that DSP of GaAs is 0.5 at a photon energy of 1.58 eV (ref. 14). The large DSP of GaAs is due to the large energy splitting in the valence band, between $P_{1/2}$ and $P_{3/2}$ bands, at the Gamma point. A theoretical calculation of DSP of transition metals will require consideration of the full band structure and energy splitting due to spin-orbit coupling.

The authors of ref. 36 theoretically calculate both IFE and OSTT contribution in metallic ferromagnets. They show that IFE can dominate over OSTT at least in Co, but they do not investigate non-magnetic metals, like Pt[36]. The authors of ref. 34 theoretically calculate the IFE-driven magnetization of several non-magnetic and ferromagnetic metals. They show that the IFE-driven magnetization is several times larger in Au and Pt than in Co, Fe and Ni due to larger spin-orbit coupling[34].

In our analysis, light absorption by dipole transitions is the fundamental mechanism for OSTT but not for IFE (for IFE, light absorption is considered only to calculate the decay of $E$-field through sample thickness). However, recent theories for IFE predict that light absorption plays an important role to induce spin and orbital magnetization[34,36]. We argue that IFE should be treated as $B$-field for short optical pulse because the IFE-induced magnetization is an equilibrium property. However, if IFE can generate magnetization on the time scale of the pulse duration, IFE can contribute to $m_{sp}$.

We compare the initial $M_z$ with theoretical IFE-induced magnetization ($m_{IFE}$) in Pt assuming timescale for $m_{IFE}$ in Pt is

shorter than the pulse duration. At $I_0 = 10^{13}\,\mathrm{W\,m^{-2}}$ and $\hbar\omega = 1.58\,\mathrm{eV}$, $m_{\mathrm{IFE}}$ in Pt are approximately 40 and 400 A m$^{-1}$, respectively, for spin and orbital (sign is opposite for spin and orbital magnetization at given light helicity)[34]. When all spin and orbital $m_{\mathrm{IFE}}$ of Pt is transferred to Co magnetization ($m_{\mathrm{Co}}$), which is mostly spin magnetization, during the pulse duration, the initial $m_{\mathrm{Co}}$ along the $z$-direction can be related with $m_{\mathrm{IFE}}$ by, $m_{\mathrm{Co}}d_{\mathrm{Co}} = m_{\mathrm{IFE}}d_{\mathrm{Pt}}$, where $d_{\mathrm{Co}}$ and $d_{\mathrm{Pt}}$ are thickness of Co and Pt layers. Then $m_{\mathrm{Co}}$ is estimated to be 8 and 80 A m$^{-1}$, respectively, by spin and orbital $m_{\mathrm{IFE}}$ of Pt with the Co(10)/Pt(2) sample, and it increases twice with the Co(10)/Pt(4) sample. The estimated $m_{\mathrm{Co}}$ by the orbital $m_{\mathrm{IFE}}$ of Pt is close to experimental observation. Note that this estimation is based on two assumptions: IFE can induce magnetization in Pt on a timescale of < 1 ps; the orbital magnetization of Pt can be transferred to the spin magnetization of Co on a timescale of < 1 ps. The orbital magnetization occurs during the dipole transition as well, but its effect on OSTT is often ignored for semiconductors[18]. Recent theory concludes that orbital magnetization has a negligible effect on magnetization dynamics of metallic ferromagnet[36].

Another consideration for the $M_z$ tiling is a spin relaxation. The light-induced spin polarization can relax to the environment before applying a torque on magnetization of FM when the spin relaxation time ($\tau_s$) is short enough. The time scale of $\tau_s$ can be estimated from spin relaxation length ($l_s$) using $\tau_s = l_s^2/D$, where $D$ is electronic diffusivity. The reported $l_s$ of Pt has a wide range $1 \sim 10$ nm, but it is related with electrical conductivity ($\sigma$)[42]. Considering $\sigma = 7 \times 10^6\,\Omega^{-1}\,\mathrm{m}^{-1}$ of our Pt film, we estimate $l_s \approx 5$ nm. With $l_s = 5$ nm and $D = 200\,\mathrm{nm^2\,ps^{-1}}$, obtained from $\sigma$, the spin relaxation time in Pt is $\tau_s \approx 0.1$ ps. The time scale for spin transfer torque ($\tau_{\mathrm{stt}}$) in the Co/Pt bilayer can be estimated from $\tau_{\mathrm{stt}} \approx l_{\mathrm{tr}}/v_F$, where $l_{\mathrm{tr}}$ is the travel length from Pt to Co, and $v_F$ is the Fermi velocity of Pt. Considering $l_{\mathrm{tr}} \approx 2$ nm, $\tau_{\mathrm{stt}}$ would be a few femtoseconds. When $\tau_{\mathrm{stt}} \ll \tau_s$, spin relaxation is not important. In addition, the spin relaxation should lead to saturation of $m_{\mathrm{sp}}$ with Pt thickness, but we do not see a saturation in the initial $M_z/M$ tilting up to Pt thickness of 4 nm.

**Ultrafast demagnetization.** The light pulse not only changes the direction of magnetization but also reduces the magnitude of magnetization via ultrafast demagnetization. The peak demagnetization, $|\Delta M|/M$, is 0.04, 0.04 and 0.25 for Co(10)/Pt(2), Fe(10)/Pt(2) and Ni(10)/Pt(2) samples, respectively (Supplementary Note 8). Ultrafast demagnetization is a result of energy transfer from light to magnetization and is related to the temperature excursion[1,43,44]. The temperature excursion per pulse is $\Delta T = F_{\mathrm{abs}}/(C_{\mathrm{FM}}d_{\mathrm{FM}} + C_{\mathrm{cap}}d_{\mathrm{cap}}) \approx 140$ K, where $C_{\mathrm{FM}}$ and $C_{\mathrm{cap}}$ is heat capacity of FM and capping layer, respectively, and $d_{\mathrm{FM}}$ and $d_{\mathrm{cap}}$ is thickness of FM and capping layer, respectively. ($\Delta T$ of magnons can exceed 140 K during the pump pulse due to non-equilibrium between electrons, phonons and magnons[1,43,44].) The more significant $|\Delta M|/M$ of Ni compared to Co or Fe is due to the relatively low Curie temperature of Ni[43,44]. We acknowledge that there are other theories for ultrafast demagnetization such as superdiffusive model[45]. According to the superdiffusive model, the larger demagnetization of Ni would be because spin dependence on electronic transport is more significant in Ni[45]. It has been proposed that IFE can lead to AO-HDS when the peak $\Delta T$ approaches the Curie temperature[11]. Recently, the possibility of AO-HDS by the combination of IFE and demagnetization was shown by simulation[46]. In this respect, a material with low Curie temperature is desirable for AO-HDS. The authors of refs 11,46 did not, however, consider OSTT.

**Conclusions.** We report the vectorial measurements of magnetization dynamics driven by helicity of light. We interpret the results in terms of two orthogonal torques: a field-like torque generated by an optomagnetic field (IFE); and a spin-transfer torque generated by a spin polarization (OSTT). We find that IFE dominates inside ferromagnetic layers, and OSTT mostly comes from a Pt capping layer. Despite our interpretation, it is possible that IFE causes a similar effect on Pt as OSTT does when the timescale for the IFE-induced magnetization is shorter than the pulse duration. Our findings present an important step towards understanding the coupling of angular momentum of light and magnetization. In particular, the capping layer dependence on OSTT and the material dependence on demagnetization can be useful in the design for materials for AO-HDS in metallic ferromagnets.

## Methods

**Pump-probe measurement.** The centre wavelength of pump and probe is 784 nm. The full-width-at-half-maximum (FWHM) of time-correlation of pump and probe is 1.15 ps, which is mostly due to FWHM of pump as it gets broaden by the large dispersion of the electro-optic modulator: we estimate FWHM of 1.1 and 0.2 ps for pump and probe, respectively. The zero time delay ($t = 0$ ps) is set to the centre of pump pulse.

**Noise suppression.** We suppress noise level using synchronous detection using a high modulation frequency (10 MHz) combined with balanced detection. In our set-up, the noise level for polar MOKE detection is on the order of 0.1 µrad per $\sqrt{\mathrm{Hz}}$ which corresponds to a fractional change of magnetization of about $10^{-5}$ for Co. We can further reduce noise by averaging multiple measurements.

**Data availability.** The data that support the findings of this study are available from the corresponding author on request.

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

## Acknowledgements

G.-M.C. was supported by the KIST Institutional Program (2E26380) and the National Research Council of Science & Technology (NST) grant (No. CAP-16-01-KIST) by the Korea government (MSIP). G.-M.C. acknowledge B.-C. Min (KIST) and K.-J. Lee (Korea University) for discussion. D.G.C. was supported by the Army Research Office MURI W911NF-14-1-0016.

## Author contributions

Sample growth and MOKE measurements were carried out at KIST by G.-M.C. with help of D.G.C.; initial experiments on MOKE measurements were carried out at University of Illinois by G.-M.C and D.G.C.; theoretical investigation for OSTT was carried out by A.S.; data analysis was carried out by G.-M.C. with help of D.G.C and A.S.; all authors discussed the results and wrote the manuscript.

## Additional information

**Competing interests:** The authors declare no competing financial interests.

