## [Peer Review File · Nature Communications]

Reviewers' comments:

Reviewer #1 (Remarks to the Author):

The manuscript "Optical-helicity-driven magnetization dynamics in metallic ferromagnets" by Choi et al. reports about a time-resolved magneto-optical study of the magnetization dynamics of the 3d-ferromagnetic metals Co, Fe and Ni. In particular, the main goal of the authors is to understand the role played by the light helicity in the dynamical response of the magnetization to optical excitation.

The question that the authors would like to answer is formulated at the beginning of the introduction "by what mechanism and to what degree can angular momentum of light be transferred to the magnetization of a metallic ferromagnet?". This is a long-standing question in the community of femtosecond magnetism, whose answer would tremendously help in further understanding of the ultrafast demagnetization process as well as of the more recently discovered all-optical magnetization switching.

Despite the broad and clear interest of the topic of this manuscript, I personally do not recommend its publication in Nature Communications in its present form. In the following I summarize my main points of criticism.

1) The authors introduce two main helicity-dependent mechanisms that will influence the magnetization dynamics: the optical spin transfer torque (OSTT) and the inverse Faraday effect (IFE).

The OSTT has been discussed up to now mainly for inorganic semiconductors. In these systems it is a very well-known fact that circularly polarized light generates a transient spin-polarization (due to the optical selection rules and the presence of spin-orbit coupling). In contrast, in a 3d-ferromagnet optical excitation will basically excite hot-electrons above the Fermi energy. These electrons will possess a spin-polarization that is in general aligned with the magnetization of the sample, in this case the x-direction (according to the reference frame defined in Fig. 1). The additional spin polarization generated by the circularity of the light in the z-direction due to the optical selection rules, will probably be negligible (if at all different than zero!) compared to the spin polarization of the optically excited carriers along the x-direction.

I am not sure about how such a small fraction of spin-polarized carriers could lead to any measurable effect: in comparison, the spin-polarization that can be generated by optical orientation in GaAs is as big as 50%.

2) Let's now assume that a small spin polarization (denoted with m_z by the authors) can be generated in the z-direction (the direction of light propagation in Fig. 1) using circularly polarized light.

The authors explain that while the spin polarization m_z should cause a torque on the sample magnetization in the z-direction, the IFE field (called B_z , and also directed along the z-direction!) will cause a torque along the y direction. I do not really understand this point. As far as I understand, in both equations describing the precession of the magnetization due to either the IFE or the OSTT, dM/dt is proportional to the cross product of M with either m_z or B_z .

So why should the OSTT-related torque point along the z-direction, while the IFE-related torque along the y-direction?

 To clarify this point, the authors should include the equations for the OSTT and the IFE in the manuscript and explain the origin of the difference in the direction of the OFE- and the OFTT-torques.

3) The authors use the polar MOKE effect to detect the z-component (in the reference frame

defined in Fig. 1 of the manuscript) of the magnetization as a function of pump-probe delay. In the MOKE measurements, the external magnetic field is aligned along the axis of the magneto-crystalline anisotropy, i.e. the x-direction.

However, as far as I understand, in order to disentangle the IFE from the OSTT, it is necessary to access BOTH the z- and the y-components of the magnetization during the time-dependent precession.

 This is probably the weakest point of the manuscript. The authors should provide also the transient values of M_y and M_x , i.e. the transient vector-reconstruction of the magnetization vector. I am not convinced that any conclusion can be driven by just measuring the z-component of M .

4) In line 94 of the manuscript, the authors write that the dynamics dominated by IFE would give a phase $\phi=90^\circ$ in the measured M_z component. To my understanding, if M is aligned along the x-direction and the IFE field B_z is along the z-direction, the IFE-related precession of magnetization will be in the xy-plane. So why should there be an oscillating IFE-related component of M along the z-direction?

 This is a critical point for the understanding of the manuscript and should be explained in much more detail.

Maybe it would be helpful to add a schematic figure showing the precession trajectories of the magnetization induced by the OSTT and by the IFE.

5) Figure 2, c. The y-Axis label is missing. What did the author plot in this graph? How can the authors conclude that the signal at $\tau=0$ derives from the OSTT, while the signal at $\tau=20\text{ps}$ derives from the IFE?

6) Instead of evaluating the DSP responsible for the OSTT, it would be helpful if the authors could reference the spin polarization induced by the circularly polarized light to the spin polarization of the 3d-ferromagnetic metals at the Fermi level. Then, it would be immediately clear how big this effect really is.

Reviewer #2 (Remarks to the Author):

The manuscript by G.M. Choi et al. reports on magnetization dynamics triggered by circularly polarized optical laser in thin ferromagnetic layers (Co, Fe and Ni). Experimentally the magnetization dynamics is probed by measuring the polar MOKE signal, from which the authors extract the system coupled dynamic. The dynamic is explained by the combination of two helicity-dependent mechanisms; first, the magnetic torque induced by an optomagnetic field created by the circularly polarized laser (inverse Faraday effect, -IFE), and second, the torque induced by spin polarized currents created through laser-induced dipole transition in systems with non-vanishing spin-orbit coupling (optical spin transfer torque, -OSTT). Although both mechanisms relies on the transfer of angular momentum from the light to the material, the IFE induces a torque perpendicular to both, the magnetization direction and the angular momentum of photons, while the torque induced by the OSTT is parallel to the angular momentum of photons. To disentangle the partial contribution of each mechanism the authors use different capping layers (Au, Pt or MgO), where the contribution of the inverse Faraday effect remains constant while the effect of optical spin transfer torque increases significantly, and fit the helicity-dependent data to a damped cosine function. Similar magnetization dynamics is observed for Co, Ni and Fe, with differences only due to the magnitude of the magnetization. A study of the ultrafast linearly polarized laser-induced demagnetization is also carried out.

Despite this work is focused on gaining understanding about the transfer of angular momentum of light to metals, a developing field which might have key implications in the technological development of magnetic recording media, and that the experimental findings are interesting, there are several questions that arise regarding the theoretical interpretation of the results that make it unsatisfactory for publication at the current state.

1) The theoretical interpretation of two helicity-dependent mechanisms triggering two orthogonal torques on the system magnetization is well presented. It is based on the initial assumption that IFE does not require absorption while OSTT is an adsorption-dependent phenomena. This assumption, if correct, would support the experimental findings. However, this is a daring assumption under the current theory development, where as pointed by the authors "rigorous theory for IFE especially in metals is still under development". Thus, for instance, two very recent theoretical papers by different groups, A. Qaiumzadeh et al. (<https://arxiv.org/pdf/1602.08305.pdf>) and M. Berritta et al. (<https://arxiv.org/pdf/1604.01188.pdf>), claim that the IFE is not an absorption-free phenomena, but rather that absorption plays a fundamental role to explain the different dynamics in different ferromagnetic metals. As a consequence, and until further theoretical clarification, the interpretation of the results of this manuscript is questionable.

Additionally, the following conclusion made by the authors, "We interpret IFE and OSTT as coming from non-absorbing and absorbing part of light, respectively", involves a false reasoning, since the analysis of the results is precisely made under such assumption, and clearly leaves not space for other possibility.

2) Despite the extensive literature of optical orientation in semiconductors, and more recently in magnetic semiconductors (leading to optical spin transfer torque), to the best of my knowledge this is the first time it is used to explain magnetization dynamics in ferromagnetic metals. The carriers spin polarization generated by the optical orientation is determined by the selection rules of the material related to the crystal symmetry and depend ultimately on an interplay between angular momentum conservation and spin-orbit interactions. Thus, in a material with vanishing spin-orbit coupling there is not spin polarization of the carries due to optical orientation (Optical Orientation, edited by F. Meier and B. P. Zakharchenya (North Holland, New York, 1984)). Hence, crystal symmetry and spin-orbit coupling play an essential role to generate spin currents due to optical orientation, which subsequently induce spin transfer torque in magnetic materials. However, from literature it is not clear which is the influence of the strength of the spin-orbit coupling in the degree of spin polarization in metals (in atoms the polarization does not depend on the strength of the spin-orbit interaction -see for instance Chapter 7 in "Optical Orientation" cited above). Therefore, it is oversimplifying to consider that "the degree of spin polarization for Pt is 25 times larger than for Co due to the stronger spin-orbit coupling" as the authors claim. In addition, and under the author's conclusion that a stronger spin-orbit coupling leads to a larger spin polarization, (and being the spin orbit strength in Au of the same magnitude than in Pt), it would be expected a spin polarization in Au about 30 times smaller than in Pt (30 times smaller energy absorption). According to the results provided by the authors, this would unambiguously lead to a negligible spin polarization in Co, but not in Au (the spin polarization in Au should be 1.2×10^{-3} and 0.2×10^{-4} for Au and Co, respectively). These results would definitely be of very strong interest if first, they were well interpreted with a further support/justification of the interpretation, and second, an analysis of the influence of the crystal structures were made.

3) The authors wrongly assume that reference [33] provides a theory to IFE in terms of an interaction Hamiltonian coupling angular momentum of light with the material. Actually, the theory developed in reference [33] contributes to the IFE as a part of it, but not providing the whole effect, whose theory can be found for instance in reference [32]. In fact, one of the conclusion of reference [33] is that the proposed interaction mechanism has a very small effect on the magnetization, which would support the small optomagnetic field obtained with the use of that theory in this manuscript. Here it is important to remark again that this is just a very small

contribution to the total IFE.

4) Another source of misunderstanding is the use of the optomagnetic field induced by the IFE as an effective Zeeman field, and its subsequent use in the Landau-Lifshitz-Gilbert equation. This leads to a wrong description of the magnetization dynamics. Contrarily, the IFE has to be treated as an induced magnetization rather than as an effective optomagnetic field, due to the fact that the laser induces different spin and orbital magnetization dynamics (<https://arxiv.org/pdf/1604.01188.pdf>).

5) As long as the IFE effect is an absorption free phenomena (assumption made by the authors in the manuscript), IFE can be related to Faraday effect, and the theoretical description made in the manuscript is valid. However, if assuming that the IFE is not an absorption free phenomena, the clear relation between both mechanisms is lost, and a new theoretical description would be needed (see point (1) above).

6) In my understanding it is not clear why the M_y component changes when changing the capping layer and its thicknesses (IFE is effectively independent to them). I would expect that as long as the IFE is effectively applying a torque into the system a M_y component should be present. This does not seem to occur for the Co/NM systems, where the M_y component goes from -2×10^{-4} to zero. Could you clarify these results?

7) In reference [40] Choi et al. justify the generation of spin currents due to the formation of a thermal gradient inside of a ferromagnetic material. It would also be expected to find such thermal gradient under the conditions of the experiments carried out in this work. Thus, it is clear that the thermal gradient-induced spin currents does not exert any torque in the magnetization of the system as long as both have the same orientation. However, when introducing a capping layer. these currents can travel through it and, upon reflection, come back to the ferromagnetic material having a different orientation than the system magnetization. This is due to the fact that the system magnetization has been under the effect of the torque induced by IFE or OSTT. Therefore, this effect should be mention and analyzed to have a complete description of the magnetization dynamics.

8) The authors also report the magnetization dynamics of the different systems with linearly polarized pump-probe experiments. The authors justify the demagnetization due to magnon heating, without considering other possible demagnetization mechanisms. Even though I am not certain how this analysis contributes to the problem studied in this work, if the authors still consider important to mention the demagnetization dynamics, they would have to justify the choice of the mechanism and argue why other possible demagnetization mechanisms can be neglected.

Smaller remarks:

9) Along the reading of the manuscript one of the main experimental findings is repeated, namely, that using Pt layer significantly enhances OSTT against Au or Mg, even before the results have been shown, and with not citation to those. This leaves the sentences unjustified and out of context, and could be interpreted as an assumption or as a known fact. Therefore it should be reformulated in a different way.

10) At the end of the manuscript the authors write that in reference [11] it has been proposed that, IFE or OSTT can lead to AO-HDS. I would like to remark that in such reference there is not mention of OSTT.

11) From the figures is very difficult to extract the details given in the manuscript. Clearer figures with, for instance, a guide line for the eye would be very helpful. Especially a vertical line at zero

time, from which the delay phase could be more easily seen.

Reviewer #3 (Remarks to the Author):

This is potentially a very important work that identifies out-of-plane and in-plane torque components generated by circularly polarized laser pulses in ferromagnetic transition metal films. By a combination of experimental and theoretical analysis the authors associate the torque oriented in the plane of the film with the inverse Faraday effect in a convincing way. The torque oriented out of the plane shows a much stronger sensitivity to the capping layers which provides an additional indication that it is of a different microscopic physics origin. The authors ascribe the out-of-plane torque to the optical spin transfer torque mechanism recently discovered in a ferromagnetic semiconductor GaMnAs. As the authors of the present paper emphasize, this mechanism is a combination of the optical spin orientation effect and of the spin transfer torque effect. Optical spin orientation is a well established field in semiconductors. In the present manuscript the authors consider optical spin polarization in Co and also in Pt. They also argue that a larger degree of optical spin polarization in Pt than in Au is due to the larger spin-orbit coupling in Pt. From the text it is however not clear what the authors assume is the microscopic physics of the optical orientation in the considered transition metal films. While the optical spin transfer torque interpretation is certainly appealing, a reference or at least a qualitative explanation of the optical orientation process in the transition metals would be desirable for making the whole story of the paper fully convincing and for increasing the impact of this interesting work. With (at least) a qualitative explanation of this the paper would be suitable for publication in Nature Communications.

Response letter to Referee's comments

Manuscript NCOMMS-16-14369-T

“Optical-helicity-driven magnetization dynamics in metallic ferromagnets” by G. M. Choi *et al.*

We thank the three referees for their positive and constructive comments. We believe our manuscript is substantially improved as a result. Below, we summarize the major changes and provide point-by-point responses to the referee's comments and suggestions that require a response. The corresponding corrections are incorporated in the revised manuscript.

Summary of major changes

1. Reviewer 1 asks us to write an equation for OSTT. We included the magnetization torque equation for OSTT as equation (1) on the first paragraphs on page five of the revised manuscript.
2. Reviewer 1 asks us to measure the y- and x-component of magnetization dynamics in addition to z-component. We include our measurements of the y-component in figure 3 of the revised manuscript and the x-component in Supplementary Note 4.
3. Reviewer 1 asks us to cite references that are relevant to OSTT in metals. We added two theoretical papers (Berrita *et al.* arXiv:1604.01188v1 and Freimuth *et al.*

arXiv:1608.02656v1) as references 33 and 35 and discuss their results on the second paragraphs on page ten of the revised manuscript.

4. Reviewer 2 asks us to consider recent theory for IFE that claims light absorption plays an important role, and light-induced magnetization is more appropriate than light-induced magnetic field for IFE. We added two theoretical papers (Berrita *et al.* arXiv:1604.01188v1 and Qaiumzadeh *et al.* arXiv:1602.08305v2) as references 33 and 34 and discuss their results on the first paragraphs on page eleven of the revised manuscript.
5. Reviewer 2 asks us to justify our description of OSTT by considering the dependence on crystal structure and spin-orbit coupling effects. This request requires full calculation of band structure, which depends on crystal structure, and energy splitting in multiple bands by spin-orbit coupling. However, the focus of our work is experimental and a full theoretical calculation is beyond the scope of our work. Instead, as our response to this comment of reviewer 1, we discuss recent theoretical papers that are relevant to OSTT in metals.
6. Reviewer 3 asks us to cite references that are relevant to OSTT in metals. This is the same comment of Reviewer 1 above.
7. Reviewer 3 asks us to clarify our assumptions for determining the degree of spin polarization (DSP) for OSTT. We describe our assumptions for the determination of DSP and supporting arguments on the first and second paragraphs on page eleven of the revised manuscript.

Details of responses

Reviewers' comments:

Reviewer #1 (Remarks to the Author):

The manuscript "Optical-helicity-driven magnetization dynamics in metallic ferromagnets" by Choi et al. reports about a time-resolved magneto-optical study of the magnetization dynamics of the 3d-ferromagnetic metals Co, Fe and Ni. In particular, the main goal of the authors is to understand the role played by the light helicity in the dynamical response of the magnetization to optical excitation.

The question that the authors would like to answer is formulated at the beginning of the introduction "by what mechanism and to what degree can angular momentum of light be transferred to the magnetization of a metallic ferromagnet?". This is a long-standing question in the community of femtosecond magnetism, whose answer would tremendously help in further understanding of the ultrafast demagnetization process as well as of the more recently discovered all-optical magnetization switching.

Despite the broad and clear interest of the topic of this manuscript, I personally do not recommend its publication in Nature Communications in its present form. In the following I summarize my main points of criticism.

1) The authors introduce two main helicity-dependent mechanisms that will influence the magnetization dynamics: the optical spin transfer torque (OSTT) and the inverse Faraday effect (IFE).

The OSTT has been discussed up to now mainly for inorganic semiconductors. In these systems

it is a very well-known fact that circularly polarized light generates a transient spin-polarization (due to the optical selection rules and the presence of spin-orbit coupling). In contrast, in a 3d-ferromagnet optical excitation will basically excite hot-electrons above the Fermi energy. These electrons will possess a spin-polarization that is in general aligned with the magnetization of the sample, in this case the x-direction (according to the reference frame defined in Fig. 1). The additional spin polarization generated by the circularity of the light in the z-direction due to the optical selection rules, will probably be negligible (if at all different than zero!) compared to the spin polarization of the optically excited carriers along the x-direction.

I am not sure about how such a small fraction of spin-polarized carriers could lead to any measurable effect: in comparison, the spin-polarization that can be generated by optical orientation in GaAs is as big as 50%.

→ Response: In our experiments, we align magnetization and light propagation directions orthogonal to each other. When magnetization and light propagation directions are parallel, spin-polarization from spin-dependent DOS probably dominates, and spin polarization from optical helicity will be overwhelmed. However, when magnetization and light propagation directions are orthogonal, spin polarization from optical helicity can be distinguished from spin polarization that results from spin-dependent DOS. We included this explanation in Supplementary Note 4.

We are indeed measuring a small quantity. We are able to measure such a small effect because we suppress noise level using synchronous detection using a high modulation frequency (10 MHz) combined with balanced detection. In our set-up, the noise level for polar MOKE detection is on the order of 0.1 μrad per root Hz which corresponds to a fractional change of magnetization of about 10^{-5} for Co when measured on a 1 sec time scale. We can further reduce

noise by averaging multiple measurements. We included this explanation in the method section of the revised manuscript.

From our analysis for OSTT, we found that degree of spin polarization of Pt is 0.03, which is about one order of magnitude smaller than that of GaAs. It is small but not negligible. (In the revised manuscript, we changed our estimate of the DSP of Pt from 0.036 to 0.03. DSP of 0.036 was obtained assuming that all M_z tilting of Co(10)/Pt(4) sample is due to spin polarization of Pt. DSP of 0.03 was obtained assuming that difference of M_z tilting between Co(10)/Pt(4) and Co(10)/Au(2) is due to spin polarization of Pt.) We included the comparison of DSP between Pt and GaAs on the first paragraph on page ten of the revised manuscript.

2) Let's now assume that a small spin polarization (denoted with m_z by the authors) can be generated in the z-direction (the direction of light propagation in Fig. 1) using circularly polarized light.

The authors explain that while the spin polarization m_z should cause a torque on the sample magnetization in the z-direction, the IFE field (called B_z , and also directed along the z-direction!) will cause a torque along the y direction. I do not really understand this point. As far as I understand, in both equations describing the precession of the magnetization due to either the IFE or the OSTT, dM/dt is proportional to the cross product of M with either m_z or B_z .

So why should the OSTT-related torque point along the z-direction, while the IFE-related torque along the y-direction?

 To clarify this point, the authors should include the equations for the OSTT and the IFE in

the manuscript and explain the origin of the difference in the direction of the OFE- and the OFTT-torques.

→ Response: The OSTT-driven torque is not simply the cross product of M with m_z . According to the theory of spin transfer torque, the direction of spin transfer torque is $-M \times (M \times I_S)$, where I_S is the spin current (Ref. 19-21 in the revised manuscript). OSTT replaces I_S with the optically-induced spin polarization, m_z ; the direction of OSTT is $-M \times (M \times \frac{dm_z}{dt})$ (Ref. 17 in the revised manuscript). Intuitively, when spin, generated either by current injection or light injection, is quickly absorbed by local magnetization of ferromagnet, the magnetization of ferromagnet should rotate toward the direction of spin by conservation of total angular momentum. It is known that spin traveling through metallic ferromagnet is absorbed by magnetization of ferromagnet on length scale of a few atomic layer (Ref. 21 in the revised manuscript). We include the equation of torque by B_z and m_z as equation (1) on the first paragraph on page five of the revised manuscript.

3) The authors use the polar MOKE effect to detect the z -component (in the reference frame defined in Fig. 1 of the manuscript) of the magnetization as a function of pump-probe delay. In the MOKE measurements, the external magnetic field is aligned along the axis of the magneto-crystalline anisotropy, i.e. the x -direction.

However, as far as I understand, in order to disentangle the IFE from the OSTT, it is necessary to access BOTH the z - and the y -components of the magnetization during the time-dependent precession.

→ Response: In the revised manuscript, we have added data for M_y and M_x dynamics measured using longitudinal MOKE. For M_y dynamics, the probe beam is tilted from the z-direction with the plane of incidence of the y-z plane. For M_x dynamics, the probe beam is tilted from the z-direction with the plane of incidence of the x-z plane. The Kerr rotation in this geometry has contributions both from M_z and M_y with incidence angle in the y-z plane, and from M_z and M_x with incident angle in the x-z plane. M_z contribution can be subtracted because it does not change sign with the sign change of external magnetic field, which is along the x-direction, while M_y (or M_x) contribution does. We included the details of longitudinal MOKE in the Supplementary Note 4. M_y dynamics is 90 degree out phase with M_z dynamics. M_x dynamics show no helicity dependence. We include results of M_y dynamics in Fig. 3 of the revised manuscript and M_x dynamics in the Supplementary Note 4.

4) In line 94 of the manuscript, the authors write that the dynamics dominated by IFE would give a phase $\phi=90$ {degree sign} in the measured M_z component. To my understanding, if M is aligned along the x-direction and the IFE field B_z is along the z-direction, the IFE-related precession of magnetization will be in the xy-plane. So why should there be an oscillating IFE-related component of M along the z-direction?

→ Response: When magnetization is tilted from the equilibrium position by optical pulse, its dynamics after the optical pulse is governed by the effective B-field (determined by shape anisotropy, crystalline anisotropy, and external magnetic field) along the equilibrium position. Since the effective B-field is along the x-direction, the magnetization dynamics is the precessional motion in the y-z plane with the center axis of the x-direction. Because of the shape

anisotropy of ferromagnets, the precessional motion is elliptical not circular in the y-z plane. After many precessional motion, the magnetization settles to the equilibrium direction (x-direction) by damping. We included this explanation on the first paragraph on page five of the revised manuscript.

5) Figure 2, c. The y-Axis label is missing. What did the author plot in this graph? How can the authors conclude that the signal at $\tau=0$ derives from the OSTT, while the signal at $\tau=20$ ps derives from the IFE?

→ Response: The phase delay, ϕ , of M_z dynamics depends on the direction of the initial tilting of magnetization. When the initial tilting of magnetization is along the z-direction, ϕ should be 0° . When the initial tilting of magnetization is along the y-direction, ϕ should be 90° . The large ϕ of 65° of the Co(10)/Au(2) sample suggests that the initial tilting of magnetization is closer to the y-direction than z-direction. Saying that M_z dynamics reaches maximum at $t \sim 20$ ps is equivalent to saying that the phase delay of M_z dynamics is 65° . To strengthen our arguments, we measure both M_z and M_y dynamics and analyze the phase delay to obtain OSTT and IFE contributions in Fig. 3 of the revised manuscript.

6) Instead of evaluating the DSP responsible for the OSTT, it would be helpful if the authors could reference the spin polarization induced by the circularly polarized light to the spin polarization of the 3d-ferromagnetic metals at the Fermi level. Then, it would be immediately clear how big this effect really is.

→ Response: In our experiment, OSTT is mostly coming from Pt. As far as we know, there is no directly relevant prior study of DSP of Pt. A related paper is Freimuth's (Freimuth *et al.* arXiv:1608.02656v1). They theoretically calculate both IFE and OSTT in metallic ferromagnets. However, they do not report results for Pt. Another related reference is Berritta's paper (Berritta *et al.* arXiv:1604.01188v1). In this paper, they calculate IFE-induced magnetization (M_{IFE}). The helicity-dependent part of M_{IFE} is several times larger in Pt and Au than in Co, Fe, Ni (see table I of Berritta's paper). Berritta *et al.* interpreted this result as being due to larger spin-orbit coupling in Au and Pt. We cite Freimuth's paper and Berritta's paper as reference [35] and [33], respectively, and discuss their results on the second paragraph on page ten of the revised manuscript.

Reviewer #2 (Remarks to the Author):

The manuscript by G.M. Choi et al. reports on magnetization dynamics triggered by circularly polarized optical laser in thin ferromagnetic layers (Co, Fe and Ni). Experimentally the magnetization dynamics is probed by measuring the polar MOKE signal, from which the authors extract the system coupled dynamic. The dynamic is explained by the combination of two helicity-dependent mechanisms; first, the magnetic torque induced by an optomagnetic field created by the circularly polarized laser (inverse Faraday effect, -IFE), and second, the torque induced by spin polarized currents created through laser-induced dipole transition in systems with non-vanishing spin-orbit coupling (optical spin transfer torque, -OSTT). Although both mechanisms relies on the transfer of angular momentum from the light to the material, the IFE induces a torque perpendicular to both, the magnetization direction and the angular momentum

of photons, while the torque induced by the OSTT is parallel to the angular momentum of photons. To disentangle the partial contribution of each mechanism the authors use different capping layers (Au, Pt or MgO), where the contribution of the inverse Faraday effect remains constant while the effect of optical spin transfer torque increases significantly, and fit the helicity-dependent data to a damped cosine function. Similar magnetization dynamics is observed for Co, Ni and Fe, with differences only due to the magnitude of the magnetization. A study of the ultrafast linearly polarized laser-induced demagnetization is also carried out.

Despite this work is focused on gaining understanding about the transfer of angular momentum of light to metals, a developing field which might have key implications in the technological development of magnetic recording media, and that the experimental findings are interesting, there are several questions that arise regarding the theoretical interpretation of the results that make it unsatisfactory for publication at the current state.

1) The theoretical interpretation of two helicity-dependent mechanisms triggering two orthogonal torques on the system magnetization is well presented. It is based on the initial assumption that IFE does not require absorption while OSTT is an adsorption-dependent phenomena. This assumption, if correct, would support the experimental findings. However, this is a daring assumption under the current theory development, where as pointed by the authors "rigorous theory for IFE especially in metals is still under development". Thus, for instance, two very recent theoretical papers by different groups, A. Qaiumzadeh et al.

(<https://arxiv.org/pdf/1602.08305.pdf>) and M. Berritta et al.

(<https://arxiv.org/pdf/1604.01188.pdf>), claim that the IFE is not an absorption-free phenomena, but rather that absorption plays a fundamental role to explain the different dynamics in different

ferromagnetic metals. As a consequence, and until further theoretical clarification, the interpretation of the results of this manuscript is questionable.

Additionally, the following conclusion made by the authors, "We interpret IFE and OSTT as coming from non-absorbing and absorbing part of light, respectively", involves a false reasoning, since the analysis of the results is precisely made under such assumption, and clearly leaves not space for other possibility.

→ Response: In the IFE part of the revised manuscript, we consider the light absorption process only to calculate the attenuation of E -field through sample thickness. To clarify the limitation of our analysis, we discuss recent theories (Berrita *et al.* arXiv:1604.01188v1 and Qaiumzadeh *et al.* arXiv:1602.08305v2) that take into account the light absorption to calculate IFE-induced magnetization on the first paragraph on page eleven of the revised manuscript. We also replace the sentence "We interpret IFE and OSTT as coming from non-absorbing and absorbing part of light, respectively" to "We interpret IFE and OSTT as two mechanisms that produce torque on magnetization along the cross product of magnetization and angular momentum of light and along the angular momentum of light, respectively, during the pulse duration of circularly polarized light." in the summary part of the revised manuscript.

2) Despite the extensive literature of optical orientation in semiconductors, and more recently in magnetic semiconductors (leading to optical spin transfer torque), to the best of my knowledge this is the first time it is used to explain magnetization dynamics in ferromagnetic metals. The carriers spin polarization generated by the optical orientation is determined by the selection rules of the material related to the crystal symmetry and depend ultimately on an interplay between

angular momentum conservation and spin-orbit interactions. Thus, in a material with vanishing spin-orbit coupling there is not spin polarization of the carries due to optical orientation (Optical Orientation, edited by F. Meier and B. P. Zakharchenya (North Holland, New York, 1984)). Hence, crystal symmetry and spin-orbit coupling play an essential role to generate spin currents due to optical orientation, which subsequently induce spin transfer torque in magnetic materials. However, from literature it is not clear which is the influence of the strength of the spin-orbit coupling in the degree of spin polarization in metals (in atoms the polarization does not depend on the strength of the spin-orbit interaction -see for instance Chapter 7 in "Optical Orientation" cited above). Therefore, it is oversimplifying to consider that "the degree of spin polarization for Pt is 25 times larger than for Co due to the stronger spin-orbit coupling" as the authors claim. In addition, and under the author's conclusion that a stronger spin-orbit coupling leads to a larger spin polarization, (and being the spin orbit strength in Au of the same magnitude than in Pt), it would be expected a spin polarization in Au about 30 times smaller than in Pt (30 times smaller energy absorption). According to the results provided by the authors, this would unambiguously lead to a negligible spin polarization in Co, but not in Au (the spin polarization in Au should be 1.2×10^{-3} and 0.2×10^{-4} for Au and Co, respectively). These results would definitely be of very strong interest if first, they were well interpreted with a further support/justification of the interpretation, and second, an analysis of the influence of the crystal structures were made.

→ Response: We are working toward developing the capability to predict OSTT of metals based on density functional theory but those calculations are not yet reliable or adequately validated. The focus of our work is experimental. As stated above, we regret that we over-interpreted the data in our original submission and we have extensively revised our manuscript to be more guarded in the conclusions we draw from the data concerning microscopic mechanisms.

In the revised manuscript, we cite and discuss Berritta's paper (Berritta *et al.* arXiv:1604.01188v1). In this paper, the helicity-dependent M_{IFE} is several times larger in Pt and Au than in Co, Fe, and Ni (see table I of Berritta's paper). Berritta *et al.* interpreted this result as being due to larger spin-orbit coupling in Au and Pt. This result is consistent with our conclusion based on the experimental data that most OSTT is coming from Pt. However, Berritta's M_{IFE} is an equilibrium quantity calculated from the density matrix, while OSTT-induced magnetization is a transient quantity. Therefore, to contribute to M_z tilting the timescale for IFE to generate equilibrium magnetization should be on the order of pulse duration of pump light. We also discuss Freimuth's paper (Freimuth *et al.* arXiv:1608.02656v1). Freimuth *et al.* theoretically calculate both IFE and OSTT contribution in metallic ferromagnet. However, they do not investigate non-magnetic metal, such as Pt. We discuss these issues on the second paragraph on page ten of the revised manuscript.

Our experiments do not allow us to draw firm conclusions about the degree of spin polarization (DSP) of the ferromagnetic layer or Au. The OSTT-driven spin polarization is proportional to the product of DSP and amount of light absorption. In FM(10)/Au(2) sample, 99 % of light absorption is by the FM layer and only 1 % of light absorption is by the Au layer. As the reviewer pointed out, even if DSP of Au is the same as that of Pt, OSTT-driven spin polarization in Au would be ~20 times smaller than in Pt because the amount of light absorption in Au is ~20 times smaller. In original submission, we erroneously concluded that the degree of spin polarization from Co is larger than that of Au in Co(10)/Au(2) because M_z dynamics is similar with Co(10)/Au(2) and Co(10)/MgO(5). However, given the uncertainty of small M_z tilting of Co(10)/Au(2) and Co(10)/MgO(5), we cannot reliably determine the contribution of the Au layer

to OSTT. Therefore, we focus on DSP of Pt and have removed the analysis of DSP of the ferromagnet layer in the revised manuscript.

3) The authors wrongly assume that reference [33] provides a theory to IFE in terms of an interaction Hamiltonian coupling angular momentum of light with the material. Actually, the theory developed in reference [33] contributes to the IFE as a part of it, but not providing the whole effect, whose theory can be found for instance in reference [32]. In fact, one of the conclusion of reference [33] is that the proposed interaction mechanism has a very small effect on the magnetization, which would support the small optomagnetic field obtained with the use of that theory in this manuscript. Here it is important to remark again that this is just a very small contribution to the total IFE.

→ Response: Our reading of the literature suggests to us that the theory of IFE in metals is a topic of great current interest that is still evolving. Our experiments contribute significantly to that on-going discussion in the literature. In the revised manuscript, we expanded our discussion of these theories including the following publications:

1. R. Mondal, M. Berritta, C. Paillard, S. Singh, B. Dkhil, P. M. Oppeneer, and L. Bellaiche, Relativistic interaction Hamiltonian coupling the angular momentum of light and the electron spin, *Phys. Rev. B*, **92**, 100402(R) (2015).
2. Berrita, M., Mondal, R., Carva, K., & Oppeneer, M., *Ab Initio* theory of coherent laser-induced magnetization in metals. arXiv:1604.01188v1 (2016).
3. Qaiumzadeh, A. & Titov, M., Theory of light-induced effective magnetic field in Rashba ferromagnets. arXiv:1602.08305v2 (2016).

4. Freimuth, F., Blügel, S., & Mokrousov Y., Laser-induced torques in metallic ferromagnets, arXiv:1608.02656v1 (2016).

Mondal and co-workers estimated $B_{\text{opt}}/|E|^2 \approx (1+\chi)10^{-22} \text{ T m}^2 \text{ V}^{-2}$, where χ_e is the electrical susceptibility. Using the relation, $\epsilon_r = 1 + \chi_e$, where ϵ_r is the relative permittivity, and ϵ_r of $-16.5 + i 23.3$ for Co at wavelength of 784 nm, and $|E|^2 \approx 10^{15} \text{ V}^2 \text{ m}^{-2}$ in our case, $B_{\text{opt}} \approx 3 \times 10^{-6} \text{ T}$. They raised a possibility of much larger χ_e for Fe relating χ_e to the anomalous Hall coefficient.

Berritta and co-workers calculate the IFE-induced magnetization (M_{IFE}) in metallic ferromagnets and related it to B_{opt} . Converting their calculation with the $I_0 = 10^{14} \text{ W m}^{-2}$ to our case with $I_0 \approx 10^{13} \text{ W m}^{-2}$, the estimated B_{opt} is 3~40 T. Note that there is asymmetry for LCP and RCP because magnetization direction and light propagation direction is the same in their case while in our experiments the magnetization is orthogonal to the direction of light propagation.

Qaiumzadeh and co-workers calculate B_{opt} in metallic ferromagnets from the direct optical transition of spin-split sub-bands. Since magnetization and light propagation lie to the same direction, their results also show asymmetry for LCP and RCP. Converting their calculation with the $E_0 = 10^9 \text{ V m}^{-1}$ to our case with $E_0 \approx 10^8 \text{ V m}^{-1}$, B_{opt} is 0.02~0.2 T.

Freimuth and co-workers consider both IFE and OSTT effect on metallic ferromagnets. They shows that both IFE and OSTT depends on quasiparticle broadening (Γ) and spin-orbit interaction. With a $\Gamma = 25 \text{ meV}$ for room temperature, they predict B_{opt} of 20 mT and 1.5 mT for Co and Fe, respectively, at $I_0 \approx 10^{13} \text{ W m}^{-2}$. The result for Fe is close to our experiment but not for Co.

We include this discussion on the paragraph that starts at the bottom on page eight of the revised manuscript.

4) Another source of misunderstanding is the use of the optomagnetic field induced by the IFE as an effective Zeeman field, and its subsequent use in the Landau-Lifshitz-Gilbert equation. This leads to a wrong description of the magnetization dynamics. Contrarily, the IFE has to be treated as an induced magnetization rather than as an effective optomagnetic field, due to the fact that the laser induces different spin and orbital magnetization dynamics (<https://arxiv.org/pdf/1604.01188.pdf>).

→ Response: Indeed some recent theories (Battiato *et al. Phys. Rev. B* **89**, 014413 (2014) and Berrita *et al. arXiv:1604.01188v1*) explain IFE in terms of induced magnetization rather than optomagnetic field. To clarify the discussion, in the revised manuscript, we emphasize the key differences between IFE-driven and OSTT-driven magnetization. First, IFE-induced magnetization is derived from a second order perturbation with respect to E -field of light, while OSTT-driven magnetization is derived from a first order perturbation. Second, their relationship to the light intensity is different: $I \propto m$ for IFE; $I \propto \frac{dm}{dt}$ for OSTT. In other words, IFE-induced magnetization is an equilibrium quantity calculated from the density matrix, while OSTT-induced magnetization is a transient quantity calculated from the transition probability, which is proportional to the square of the density matrix. Note that IFE-induced B -field and magnetization can be related by $m = \chi_m B$, where χ_m is the static magnetic susceptibility (Battiato *et al. Phys. Rev. B* **89**, 014413 (2014)). However, when one examines transient magnetization dynamics due to a short optical pulse, one needs to treat IFE as a transient B -field in the LLG equation. We include this discussion on the paragraph that starts at the bottom on page three of the revised manuscript. However, we admit that when IFE can lead to equilibrium magnetization at times of pulse

duration we cannot distinguish whether light-induced magnetization is coming from IFE or OSTT. We also discuss this possibility on the first paragraph on page eleven of the revised manuscript.

5) As long as the IFE effect is an absorption free phenomena (assumption made by the authors in the manuscript), IFE can be related to Faraday effect, and the theoretical description made in the manuscript is valid. However, if assuming that the IFE is not an absorption free phenomena, the clear relation between both mechanisms is lost, and a new theoretical description would be needed (see point (1) above).

→ Response: We use the IFE theory of absorption free because we interpret M_y tilting is a result of B_{opt} which is driven by non-absorbing part of light. (We consider light absorption only to calculate attenuation of E -field through thickness). Some of recent theories (Berrita *et al.* arXiv:1604.01188v1 and Qaiumzadeh *et al.* arXiv:1602.08305v2) predict that light absorption plays an important role in generating IFE-driven magnetization. When IFE establish equilibrium magnetization at timescale of pulse duration, IFE can contribute to M_z tilting. We included this discussion on the first paragraph on page eleven of the revised manuscript.

6) In my understanding it is not clear why the M_y component changes when changing the capping layer and its thicknesses (IFE is effectively independent to them). I would expect that as long as the IFE is effectively applying a torque into the system a M_y component should be present. This does not seem to occur for the Co/NM systems, where the M_y component goes from -2×10^{-4} to zero. Could you clarify these results?

→ Response: It is because of the uncertainties of our original measurement approach. We found that analyzing just M_z dynamics on a time scale of 0 to 500 ps tends to have large errors. In preparing the revisions to the manuscript we improved the experiments and now use two approaches to reduce the uncertainties. First, we analyze average data of ten to twenty measurements on the time scale of 0 to 120 ps. Second, we analyze both M_z and M_y dynamics. Smaller uncertainties in the new data show that IFE contribution does not change with capping layer. We include data of M_z and M_y dynamics in Fig. 3 of the revised manuscript.

7) In reference [40] Choi et al. justify the generation of spin currents due to the formation of a thermal gradient inside of a ferromagnetic material. It would also be expected to find such thermal gradient under the conditions of the experiments carried out in this work. Thus, it is clear that the thermal gradient-induced spin currents does not exert any torque in the magnetization of the system as long as both have the same orientation. However, when introducing a capping layer, these currents can travel through it and, upon reflection, come back to the ferromagnetic material having a different orientation than the system magnetization. This is due to the fact that the system magnetization has been under the effect of the torque induced by IFE or OSTT. Therefore, this effect should be mention and analyzed to have a complete description of the magnetization dynamics.

→ Response: We agree that a large spin current should be generated in the x-direction, travel through the capping layer, and come back to ferromagnet. If there is any helicity-dependent effect on this process, this spin current should lead to helicity-dependent torque in the x-direction. However, we do not see any helicity dependence in M_x dynamics. We include data of M_x dynamics in Supplementary Note 4.

8) The authors also report the magnetization dynamics of the different systems with linearly polarized pump-probe experiments. The authors justify the demagnetization due to magnon heating, without considering other possible demagnetization mechanisms. Even though I am not certain how this analysis contributes to the problem studied in this work, if the authors still consider important to mention the demagnetization dynamics, they would have to justify the choice of the mechanism and argue why other possible demagnetization mechanisms can be neglected.

→ Response: We show demagnetization data to emphasize material dependence on all-optical helicity-dependent switching. To avoid any presumption about the mechanism for demagnetization, we cite another mechanism in the revised manuscript, so called superdiffusive theory (Battiato *et al. Phys. Rev. Lett.* **105**, 027203 (2010)). The superdiffusive theory explains demagnetization in terms of spin-dependent life time and velocity of hot electrons. In this view point, larger demagnetization of Ni would be a result of larger spin-dependent transport property of hot electrons. We include this discussion on the first paragraph on page twelve of the revised manuscript.

Smaller remarks:

9) Along the reading of the manuscript one of the main experimental findings is repeated, namely, that using Pt layer significantly enhances OSTT against Au or Mg, even before the results have been shown, and with not citation to those. This leaves the sentences unjustified and

out of context, and could be interpreted as an assumption or as a known fact. Therefore it should be reformulated in a different way.

→ Response: We reformulate our manuscript by starting the analysis of IFE and OSTT after we show all data and decompose the initial M_z and M_y tilting of all samples. Since the initial M_y tilting does not depend on capping materials, we interpret initial M_y tilting is due to IFE in ferromagnets. Since the initial M_z tilting is greatly enhanced with a Pt capping, we interpret initial M_z tilting is mostly due to OSTT in Pt. We indicated two assumptions for interpretation of OSTT (First, M_z tilting is due to OSTT not IFE. Second, we neglect effect of orbital polarization and spin relaxation.) on the first and second paragraphs on page eleven of the revised manuscript.

10) At the end of the manuscript the authors write that in reference [11] it has been proposed that, IFE or OSTT can lead to AO-HDS. I would like to remark that in such reference there is not mention of OSTT.

→ Response: We corrected that reference [11] only consider IFE in the revised manuscript.

11) From the figures is very difficult to extract the details given in the manuscript. Clearer figures with, for instance, a guide line for the eye would be very helpful. Especially a vertical line at zero time, from which the delay phase could be more easily seen.

→ Response: We included vertical guide line in the figures of revised manuscript at time delay of 1 ps. We choose time delay of 1 ps instead of 0 ps because the torque on magnetization completes after pump pulse. The pulse duration of pump is 1.1 ps and the zero of time delay is set to the center of pump pulse.

Reviewer #3 (Remarks to the Author):

This is potentially a very important work that identifies out-of-plane and in-plane torque components generated by circularly polarized laser pulses in ferromagnetic transition metal films. By a combination of experimental and theoretical analysis the authors associate the torque oriented in the plane of the film with the inverse Faraday effect in a convincing way. The torque oriented out of the plane shows a much stronger sensitivity to the capping layers which provides an additional indication that it is of a different microscopic physics origin. The authors ascribe the out-of-plane torque to the optical spin transfer torque mechanism recently discovered in a ferromagnetic semiconductor GaMnAs. As the authors of the present paper emphasize, this mechanism is a combination of the optical spin orientation effect and of the spin transfer torque effect. Optical spin orientation is a well established field in semiconductors. In the present manuscript the authors consider optical spin polarization in Co and also in Pt. They also argue that a larger degree of optical spin polarization in Pt than in Au is due to the larger spin-orbit coupling in Pt. From the text it is however not clear what the authors assume is the microscopic physics of the optical orientation in the considered transition metal films. While the optical spin transfer torque interpretation is certainly appealing, a reference or at least a qualitative explanation of the optical orientation process in the transition metals would be desirable for making the whole story of the paper fully convincing and for increasing the impact of this interesting work. With (at least) a qualitative explanation of this the paper would be suitable for publication in Nature Communications

→ Response: We use two assumptions for our analysis of OSTT. First we assume M_z tilting is due to OSTT not IFE. Some recent IFE theories (Battiato *et al. Phys. Rev. B* **89**, 014413 (2014) and Berrita *et al. arXiv:1604.01188v1*) claim IFE can induce magnetization. Following our response to reviewer 2's comment, we argue that, in short light pulse, IFE should be treated as a transient B -field rather than a magnetization because IFE-driven magnetization is a quantity at equilibrium. However we admit that when IFE can generate magnetization on a timescale of the pulse duration IFE can contribute to M_z . Second we determine DSP for OSTT ignoring orbital polarization and spin relaxation. The light-induced dipole transition results in both orbital and spin polarization, but the former is often ignored for semiconductor. Recent theory claims that orbital polarization has negligible effect on magnetization dynamics of metallic ferromagnet as well (Freimuth *et al. arXiv:1608.02656v1*). The spin polarization can relax to environment before applying a torque on magnetization of FM when the spin relaxation time (τ_s) is short enough. The time scale of τ_s can be estimated from spin relaxation length (l_s) using $\tau_s = \frac{l_s^2}{D}$, where D is diffusivity. The reported l_s of Pt has a wide range 1~10 nm, but it is related with electrical conductivity (σ) (Sagasta *et al. arXiv:1603.04999*). Considering $\sigma = 7 \times 10^6 \Omega^{-1} \text{ m}^{-1}$ of our Pt film, l_s would be ≈ 5 nm. With l_s of 5 nm and D of $200 \text{ nm}^2 \text{ ps}^{-1}$, obtained from σ , τ_s for Pt would be ≈ 100 fs. The time scale for spin transfer torque (τ_{stt}) in the Co/Pt bilayer can be estimated from $\tau_{\text{stt}} \approx \frac{l_{\text{tr}}}{v_F}$, where l_{tr} is the travel length from Pt to Co, and v_F is the Fermi velocity of Pt. Considering l_{tr} of ~ 2 nm, τ_{stt} would be a few femtoseconds. When $\tau_{\text{stt}} \ll \tau_s$, spin relaxation is not important. In addition, the spin relaxation should lead to saturation of m_{sp} with Pt thickness, but we do not see the saturation in the initial M_z/M tilting up to Pt thickness of 4 nm. We

included this discussion on the first and second paragraphs on page eleven of the revised manuscript.

We also discuss results of recent theories (Freimuth *et al.* arXiv 1608.02656v1 and Berritta *et al.* arXiv 1604.01188) that are relevant to OSTT in metals. This is the same as the response to reviewer 1's final comment.

Reviewers' comments:

Reviewer #1 (Remarks to the Author):

I think that the authors did a good job in replying to the quite tough questions of the three referees, especially considering that the microscopic mechanisms mediating the transfer of angular momentum between light and magnetic materials are far by being fully understood.

I am sure that the present manuscript will generate strong interest in the femtosecond magnetism community and stimulate further theoretical efforts to shed more light on the physics behind the optical spin transfer torque and inverse Faraday effect.

To my opinion the revised version of the manuscript should thus now be published without any further revision.

Reviewer #2 (Remarks to the Author):

I thank the authors for addressing and giving response to the comments raised by me and the other two referees, and I also acknowledge that in its new version the manuscript is considerably improved. However, I still cannot recommend its publication in Nature Communications. My major concern remains regarding the theoretical interpretation. Despite the authors explicitly specify that the focus of the work is experimental, they still assume that two different mechanisms for optical helicity-driven magnetization dynamics, namely inverse Faraday effect (IFE) and optical spin transfer torque (OSTT), provide orthogonal torques, and concentrate in measuring such perpendicular torques. Nonetheless, as far as I understand, this essential assumption is not yet clear and, as pointed out by the authors, it is possible that IFE can generate magnetization on the time scale of the pulse duration (1 ps) that could provide a torque in the same direction than that induced by OSTT. If that was the case, the interpretation of the results would be different, and would invalidate the conclusions drawn in the manuscript.

In addition I would like to mention that the paragraph in page 3 indicating the key differences between IFE and OSTT is not clear and does not help to understanding both mechanisms. More specifically, I do not understand the following statement: "while OSTT-induced magnetization is a transient quantity calculated from the transition probability and therefore proportional to the square of the density matrix". In my understanding, the density matrix includes the transition amplitudes and I do not see why it should be proportional to the square of the density matrix (when the density matrix is modified it carries information for both the IFE and the transient torque coming from OSTT).

Reviewer #3 (Remarks to the Author):

The authors have properly addressed my comments and I recommend the paper for publication.

Response letter to Referee's comments

Manuscript NCOMMS-16-14369A

“Optical-helicity-driven magnetization dynamics in metallic ferromagnets” by G. M. Choi *et al.*

Below, we summarize the point-by-point responses to the second referee's comments. The corresponding corrections are incorporated in the revised manuscript. All changes are highlighted with blue color in the revised manuscript.

Reviewers' comments:

Reviewer #2 (Remarks to the Author):

I thank the authors for addressing and giving response to the comments raised by me and the other two referees, and I also acknowledge that in its new version the manuscript is considerably improved. However, I still cannot recommend its publication in Nature Communications. My major concern remains regarding the theoretical interpretation. Despite the authors explicitly specify that the focus of the work is experimental, they still assume that two different mechanisms for optical helicity-driven magnetization dynamics, namely inverse Faraday effect (IFE) and optical spin transfer torque (OSTT), provide orthogonal torques, and concentrate in measuring such perpendicular torques. Nonetheless, as far as I understand, this essential

assumption is not yet clear and, as pointed out by the authors, it is possible that IFE can generate magnetization on the time scale of the pulse duration (1 ps) that could provide a torque in the same direction than that induced by OSTT. If that was the case, the interpretation of the results would be different, and would invalidate the conclusions drawn in the manuscript.

→ Response: For IFE-driven magnetization by short optical pulse, we must consider dynamic behavior. For example, the alignment of magnetization along the B -field will take a few nanoseconds for Co as magnetization undergoes a damped precessional motion to approach the equilibrium position. However, if the alignment of magnetization occurs during the pulse duration, we can treat IFE as a magnetization rather than a B -field. In our analysis, we assume that the timescale for IFE to induce magnetization is much longer than the pulse duration, and treat IFE as a transient B -field created by the optical pulse and solve the torque equation. We include this discussion on the paragraph that starts at the bottom on page three of the revised manuscript.

To consider the possible contribution of IFE to spin-transfer torque along the z -direction, we compare the initial M_z with theoretical IFE-induced magnetization (m_{IFE}) in Pt assuming the timescale for m_{IFE} in Pt is shorter than the pulse duration. At $I_0 = 10^{13} \text{ W m}^{-2}$, and $\hbar\omega = 1.58 \text{ eV}$, m_{IFE} in Pt are approximately 40 and 400 A m^{-1} , respectively, for spin and orbital (sign is opposite for spin and orbital magnetization at given light helicity) [Berrita *et al. Phys. Rev. Lett.* **117**, 137203 (2016)]. When all spin and orbital m_{IFE} of Pt is transferred to Co magnetization (m_{Co}), which is mostly spin magnetization, during the pulse duration, the initial m_{Co} along the z -direction can be related with m_{IFE} by, $m_{\text{Co}}d_{\text{Co}} = m_{\text{IFE}}d_{\text{Pt}}$, where d_{Co} and d_{Pt} are thickness of Co and Pt layers. Then m_{Co} is estimated to be 8 and 80 A m^{-1} , respectively, by spin and orbital m_{IFE}

of Pt with the Co(10)/Pt(2) sample, and it increase twice with the Co(10)/Pt(4) sample. The estimated m_{Co} by orbital m_{IFE} of Pt is close to experimental observation. Note that this estimation is based on two assumptions: IFE can induce magnetization in Pt on a timescale of <1 ps; the orbital magnetization of Pt can be transferred to the spin magnetization of Co on a timescale of <1 ps. We included this discussion on the second paragraph on page twelve of the revised manuscript.

We also included the statement of “Despite our interpretation, it is possible that IFE causes a similar effect on Pt as OSTT does when the timescale for IFE-induced magnetization is shorter than the pulse duration” in the discussion section that starts at the bottom of page thirteen of the revised manuscript.

2) In addition I would like to mention that the paragraph in page 3 indicating the key differences between IFE and OSTT is not clear and does not help to understanding both mechanisms. More specifically, I do not understand the following statement: "while OSTT-induced magnetization is a transient quantity calculated from the transition probability and therefore proportional to the square of the density matrix". In my understanding, the density matrix includes the transition amplitudes and I do not see why it should be proportional to the square of the density matrix (when the density matrix is modified it carries information for both the IFE and the transient torque coming form OSTT).

→ Response: We confused the density matrix with matrix element. We correct the statement by “while OSTT-induced magnetization derives from the rate of spin generation, calculated from

probability of interband transitions” on the paragraph that starts at the bottom on page three of the revised manuscript.

REVIEWERS' COMMENTS:

Reviewer #2 (Remarks to the Author):

I would like to thank the authors for addressing my previous criticism/questions and for taking into considerations my comments to improve the manuscript.

The new version of the manuscript addresses satisfactorily my concerns. Now, the authors not only provide novel experimental results regarding all optical helicity-dependent switching, but also give an adequate theoretical interpretation by establishing clear assumptions which give the reader the opportunity to understand the theory and its possible limitations. Hence, I recommend its publication in Nature Communications, without further comments.